# Interaction of the CMTM7 rs347134 Polymorphism with Dietary Patterns and the Risk of Obesity in Han Chinese Male Children

**DOI:** 10.3390/ijerph17051515

**Published:** 2020-02-26

**Authors:** Qi Zhu, Kun Xue, Hong Wei Guo, Fei Fei Deng, Yu Huan Yang

**Affiliations:** 1Department of Nutrition, School of Public Health, Fudan University, Shanghai 200032, China; zgjsntzq@126.com (Q.Z.);; 2Department of Medicine, Nantong University Xinglin College, Nantong 226008, Jiangsu, China; 3School of Public Health, Nantong University, Nantong 226019, Jiangsu, China

**Keywords:** CMTM7 rs347134, SNP, children obesity, dietary patterns, interaction analysis

## Abstract

A genome-wide association study (GWAS) in the Han Chinese population had found that single nucleotide polymorphism (SNP) on the CMTM7 gene rs347134 was significantly associated with Body Mass Index (BMI). In the present study, the association of the rs347134 SNP with obesity and its interaction with dietary patterns (DPs) were explored in Han Chinese children. This cross-sectional study group included 1292 children, in whom obesity-related indicators were evaluated, the rs347134 SNP was genotyped by improved Multiple Ligase Detection Reaction (iMLDR), and the DPs were identified by principal component factor analysis. The GG genotype exhibited higher odds of general overweight/obesity (*P* = 0.038) and central obesity (*P* = 0.039) than AA + GA genotypes in boys. Four DPs of boys were identified: healthy balanced (HBDP), nuts and sweets-based (NSDP), animal food-based (AFDP), and wheaten and dairy-based (WDDP). Boys with the GG genotype were significantly more inclined to AFDP (*P* = 0.028) and had a shorter sleep duration (*P* = 0.031). Significant interactions were observed; boys with the GG genotype displayed a higher LDL in AFDP (*P* = 0.031) and higher FBG in NSDP (*P* = 0.038), respectively. Our findings indicate for the first time that the GG genotype of CMTM7 rs347134 is potentially a novel obesity risk factor for Han Chinese male children and is associated with dietary patterns more or less.

## 1. Introduction

Overweight and obesity have become a major global public health concern and challenge. Almost two-thirds of the world’s obese people live in low- and middle-income countries, where children are likely to be increasingly overweight or obese [1]. Evidence shows that from 1985 to 2014, the incidence of overweight and obese children in China increased rapidly from 0.1% to 7.3% [2,3,4].

Childhood obesity has both short- and long-term health effects, including increased blood pressure, reduced glucose tolerance, dyslipidemia, metabolic syndrome, type 2 diabetes, cardiovascular disease, sleep apnea, and the risk of a variety of psychosocial problems [5,6]. In addition, obesity can lead to increased socio-economic burden.

The etiology of obesity is complex and multifactor [7]. Whether environmental factors or genetic factors, no single path leads to obesity. Sex, ethnic groups, and even different individuals have different sensitivities to these factors. The interaction between environmental changes and genetic factors can lead to a significant increase in the prevalence of obesity. [8]. It could provide new insights into the variation of obesity susceptibility among individuals [9]. These environmental factors have been widely studied and are well known, including dietary patterns, physical activity, and sleep duration. Many genes associated with obesity are also increasingly known, such as the obesity (OB) gene [10], fat mass and obesity-related (FTO) gene [11], melanocortin receptor (MC4R) gene [12], etc. [13]. Some polymorphisms in the FTO gene have been found to be associated with obesity [14], higher energy intake [15], and decreasing sleep duration [16]. The MC4R rs17782313 polymorphism has been widely studied and found to be significantly associated with obesity [17], increased snacking, and hunger [18]. Now, more and more genetic polymorphisms have been found to be associated with obesity and environmental factors. Previous studies mostly focused on populations of European descent and on adults, however subsequent studies have also attempted to identify such associations in Chinese populations and in children. For example, four gene loci of SEC16B rs543874, MC4R rs17782313, MAP2K5 rs2241423, and KCTD15 rs11084753 were found to be significantly correlated with obesity in Chinese children and adolescents after adjusting for age and sex, and had an interaction with dietary behavior [19]. Different nationalities have different ancestors and may have different genetic backgrounds [20]. There are 56 nationalities in China, more than 90% of which are from the Han nationality. Therefore, it is of great significance to study the association and interaction between environmental and genetic polymorphism factors for prevention and intervention of childhood obesity in Han Chinese populations.

A genome-wide association study (GWAS) in Han Chinese adult populations had found that the single nucleotide polymorphism (SNP) on the CMTM7 gene rs347134 is significantly associated with Body Mass Index (BMI) [21]. Whether there is a significant correlation with children remains to be verified. CMTM (CKLF-like MARVEL transmembrane domain-containing family) currently includes 9 members: chemokine-like factor and CMTM1-8, which were first reported by Chinese researchers [22,23]. CMTM7 located on chromosome 3 (3p22.3) is widely expressed in human tissues, especially in immune cells, and play an important role in immune, reproductive and hematopoietic systems, as well as participate in the occurrence and development of tumors.

However, to our knowledge so far, no studies have been carried out about the relationships between the CMTM7 rs347134 polymorphism, the odds of general overweight/obesity or central obesity, and obesity-related anthropometric, physiological, and biochemical indicators, with a concurrent research also of their possible interactions with dietary, activity, and sleep assessments. Thus, the present study was undertaken to reveal the association and interaction of the CMTM7 rs347134 polymorphism with dietary patterns, physical activity, and sleep duration, and the consequent risk of obesity in Han Chinese children.

## 2. Materials and Methods

### 2.1. Study Population

This study was subordinate to the nutrition-based comprehensive intervention study on childhood obesity in China (NISCOC): a multi-centered, randomized and cluster-controlled trial (Chinese clinical trial registry (primary registry in the WHO registry network) identifier: ChiCTR-TRC-00000402). The data were derived from the baseline survey of the Shanghai center. In order to ensure the samples were representative, a stratified random cluster sampling was carried out. Firstly, primary schools were divided into urban, suburban, and rural layers; two schools were randomly selected in each layer, and then two classes were randomly selected from each grade (1–4 grade, 7–12 years). Lastly, 30–40 pupils were drawn from the classes as the research participants. Finally, a total of 1511 healthy Han nationality children were surveyed. The study was approved by the ethical review committee of institute of Nutrition and Food Safety, Chinese Center for Disease Control and Prevention (Ethics number: 20081201). All the participating pupils, parents, or legal guardians were informed of the purpose and procedures of the study and signed the informed consent.

### 2.2. Anthropometric Assessments

Fasting body weight and height was measured by an electronic column scale (GMCS-I; Yishen, Shanghai, China) in light clothing and barefoot. The accuracy of this instrument is greater than ±0.1%, according to the manufacturer. BMI was calculated as weight (kg)/height squared (m^2^). Subjects were classified as normal, overweight, and obese according to the “Chinese school-age children and adolescents overweight and obesity screening BMI classification criteria” proposed by the China Working Group on Obesity [24]. Waist circumference (WC) was measured by a non-elastic tape midway between the lowest rib margin and iliac crest at the end of a normal exhalation, without any pressure to the body surface and to an accuracy of 0.1 cm. If the participant was obese, the measurement should be made around the fattest position. Measurements were repeated twice and the error should have been less than 2 cm; then an average was taken. The waist-to-height ratio (WHtR) was calculated as dividing WC by height; figures greater than 0.47 and 0.45 were considered as central obesity for boys and girls, respectively [25]. Blood pressure was measured using a mercury sphygmomanometer in a quiet room with at least a 10-min rest period before the measurement.

### 2.3. Dietary and Physical Activity Assessments

Homemade food record form sheets with food size labels were used to record all kinds and amounts of foods eaten by the participants for three consecutive days (including 2 school days and 1 weekend), at home and out. Subsequently, the daily energy and nutrients intake of each participant was calculated by a self-developed SY nutrition analysis and recipe formulation software. Food items were converted to 21 food groups for extracting dietary patterns.

The 7-days retrospective physical activity questionnaires were used to review the physical activities of the participants, including dynamic state duration (walking, bicycling, sports, etc.), static state duration (doing homework, playing cell phone or computer, watching TV, etc.), and sleep duration. Afterwards, the dynamic state duration was converted to the metabolic equivalent hours per week (MET-h/wk) according to Ainsworth [26].

All above surveys were conducted by trained professionals and completed under the guidance of parents or guardians.

### 2.4. Laboratory Assays

Fasting serum triglycerides (TG), total cholesterol (TC), high density lipoprotein (HDL), low density lipoprotein (LDL), and glucose (FBG) were detected from blood samples drawn early in the morning. All samples were assayed by standard methods (Enzyme Kit: Fenghui Medical Technology Company, Shanghai, China; LX-20 automatic biochemical analyzer: Beckman Clouter, CA, USA).

### 2.5. Genotyping

DNA was extracted from whole blood by the phenol-chloroform method. The CMTM7 gene rs347134 SNP was genotyped by the improved Multiple Ligase Detection Reaction (iMLDR) technique at GENESKY Biotechnology Company, Shanghai, China. PCR used the following primers: forward 5′-TGGGTGCCATCTTATCCACTGA-3′; reverse 5′-CCATCATTGCCGCTTGTGAAAT-3′ (Sangon Biotech, Shanghai, China). PCR reactions were performed in a final volume of 20 μL, containing 1 μL extracted DNA, 1 μL primers, 1* GC-I buffer (Takara, Shanghai, China), 3.0 mM Mg^2+^, 0.3 mM dNTP (Generay Biotech, Shanghai, China), and 1 U HotStarTaq polymerase (Qiagen, Dusseldorf, NW, Germany) with the following conditions in a DNA thermocycler: DNA templates were denatured at 95 °C for 2 min; amplification consisted of 11 cycles at 94 °C for 20 s, 65 °C for 40 s (−0.5 °C/cycle), 72 °C for 1.5 min, and 24 cycles at 94 °C for 20 s, 59 °C for 30 s, 72 °C for 1.5 min, with a final extension at 72 °C for 2 min. Amplified DNA(20 μL) was digested with a 5 U SAP enzyme (Promega, Madison, WI, USA) and 2 U Exonuclease I enzyme (Epicentre, Ipswich, MA, USA) at 37 °C for 1 h, and then inactivated at 75 °C for 15 min. Ligase chain reaction (LCR) used the following primers: FG: 5′-TCTCTCGGGTCAATTCGTCCTTCCTGTGCAGAAAAACAGA CTGTGTCG-3′; FA: 5′-TGTTCGTGGGCCGGATTAGTCCTGTGCAGAAAAACAGACTGTGTCA-3′; FP: 5′-TTTACATAACCATAACATGGACAGAGAATCA-3′ (Sangon Biotech, Shanghai, China). LCR reactions system contained 1 μL 10* ligation buffer, 0.25 μL high temperature ligase (Thermo Fisher Scientific, Waltham, MA, USA), 0.4 μL 5′primers (1 μM), 0.4 μL 3′primers (2 μM), 2μL purified product of PCR, and 6 μL ddH_2_O with the following conditions: 38 cycles at 94 °C for 1 min and 56 °C for 4 min. Dilute ligation product (0.5 μL) was blended with 0.5 μL Liz500 Size Standard, 9 μL Hi-Di (Applied Biosystems, Foster City, CA, USA), and denatured at 95 °C for 5 min. The sequencing process was performed using the ABI3730XL automated sequencer (Applied Biosystems, Foster City, CA, USA). The original data collected from the sequencer were analyzed with GeneMapper 4.1 (Applied Biosystems, Foster City, CA, USA).

### 2.6. Statistical Analyses

Data were analyzed using the Statistical Package for the Social Science (SPSS version 22.0, IBM, Armonk, NY, USA). The Kolmogorov–Smirnov test was used for assaying normal distributions. The independent *t*-test was used to compare difference of obesity, dietary, physical activity related indicators by sex, overweight/obesity or not, and genotype. Pearson’s *χ*^2^ statistic was applied to examine the Hardy–Weinberg equilibrium for the SNP and compare the distribution of the quartile groups of dietary patterns by genotype. Confounder factors were adjusted for by ANCOVA. Odds ratio (OR) and 95% confidence intervals (95% CI) was calculated by logistic regression. Dietary patterns were extracted by factor analysis with Varimax rotation and based on 21 food groups; they were adjusted for total energy intake by way of residual regression. Dietary patterns were retained based on criteria including a scree plot and an eigenvalue greater than 1.5. Comparing the quantitative variable between genotypes or the quartile of the dietary patterns was performed by ANOVA and Tukey’s-b post-hoc test. The interaction between genotypes and dietary patterns of quantitative and qualitative variables was analyzed by general linear and logistic regression, respectively. A *p* value < 0.05 was considered significant.

## 3. Results

After excluding null, invalid, and contaminated samples, a total of 1292 (658 (50.93%) boys and 634 (49.07%) girls) participants were included in the study. The age range was between 7 and 12 years old, and the mean (SD) of age was 9 (1.2). This survey found that the total rate of general overweight/obesity was 19.43%, with 24.17% boys and 14.51% girls, and the total rate of central obesity was 14.55%, with 16.72% boys and 12.30% girls. It was revealed that male children were more likely to be generally overweight/obese (OR 1.81; 95% CI 1.28–2.57, *P* = 0.001) and have central obesity (OR 1.50; 95% CI 1.02–2.20, *P* = 0.037) than female children.

Both general overweight/obesity and central obesity children exhibited significantly higher SP, DP, TG, and LDL but a significantly lower HDL level, compared to the normal children. The general overweight/obesity group also showed a significantly higher FBG than the normal group (Table 1). Additionally, male children exhibited a significantly higher general overweight/obesity and central obesity rate, BMI, waist, WHtR, SP, LDL, and FBG, but a significantly lower HDL level, compared to the female children (Table 2). So, the following study was stratified by sex.

### 3.1. Association between CMTM7 rs347134 Polymorphism and Obesity-Related Indices

In this study, the prevalence of the G allele was 53.29%, while for the A allele it was 46.71%. There were three genotypes: GA (47.60%), GG (29.50%), and AA (22.91%), which were in Hardy–Weinberg equilibrium (using a Chi square test, *P* = 0.232). The minor allele frequency (MAF) was 0.469 of A.

It was found that boys with the GG genotype exhibited a higher general overweight/obesity incidence (30.1%) compared to those with the AA + GA genotypes (21.7%) before (OR 1.58; 95% CI 1.08–2.53, *P* = 0.030) and after (OR 1.52; 95% CI 1.02–2.46, *P* = 0.036) adjusting for potential confounders (age, total energy intake, physical activity). Meanwhile, boys with the GG genotype were more likely to have central obesity (22.8%) than the AA + GA genotypes (14.2%) before (OR 1.73; 95% CI 1.03–2.91, *P* = 0.038) and after (OR 1.69; 95% CI 1.01–2.85, *P* = 0.039) controlling for potential confounders. In addition, boys with the GG genotype exhibited an increasing BMI, waist, WHtR, and SP compared to those with the AA + GA genotype before and after adjusting for potential confounders (Table 3). However, there were no significant differences between genotypes in girls for the abovementioned indices. For this reason, girls were not included in subsequent study.

### 3.2. Association between Dietary Patterns and Obesity-Related Indices in Male Children

According to the dietary data, four major dietary patterns of boys were identified: healthy balanced dietary patterns (HBDP), nuts and sweets-based dietary patters (NSDP), animal foods-based dietary patterns (AFDP), and wheaten foods and dairy-based dietary patterns (WDDP). HBDP was characterized by balanced consumption of various foods, such as vegetables, eggs, tubers, and dairy. The NSDP was characterized by a high consumption of nuts, pastries, and sweets. The AFDP was high in animal foods, such as pork, fish, poultry, and also some vegetables. The WDDP was characterized by a high intake of wheaten foods and some diary, but a low intake of rice (Table 4).

Boys following NSDP were associated with having higher FBG levels, both before (*P* = 0.03) and after (*P* = 0.04) adjustment for confounders (age, physical activity). The AFDP was associated with higher levels of LDL before (*P* = 0.05) and after (*P* = 0.04) adjustment. The mean value of the participant characteristics and residual adjusted measured variables, according to quartile of the four dietary patterns, are shown in Table 5.

### 3.3. Association of CMTM7 rs347134 Polymorphism with Energy Intake, Activity and Dietary Pattern in Boys

It was found that the energy intake proportion of protein was higher (*P* = 0.043), but carbohydrate was lower (*P* = 0.046) in GG genotype boys. Boys with the GG genotype also displayed a significantly shorter sleep duration (*P* = 0.031) according to independent *t*-test (Table 6). For dietary patterns, it was revealed that boys with the GG genotype were more inclined to AFDP (*P* = 0.028), but it was inversely in AA + GA according to logistic regression analysis after adjusting for age and physical activity (Figure 1). There were no significant association between the other three dietary patterns and the CMTM7 rs347134 polymorphism (not shown).

For physical activity time, sedentary time, and sleep duration, there were no significant relationship with the incidence of general overweight/obesity or central obesity and other obesity-related anthropometric, physiological, and biochemical indices (not shown).

### 3.4. Interaction of CMTM7 rs347134 Polymorphism with Energy Intake, Activity and Dietary Pattern in Boys

Through a more in-depth study, significant interactions were observed between the CMTM7 rs347134 polymorphism and NSDP in terms of FBG (*P*-interaction = 0.030), in such a way that the last quartile of the NSDP compared to the first quartile increased the terms of FBG with the GG genotype (*P* = 0.031) while reducing them for those carrying the AA + GA genotypes (*P* = 0.040). Significant interactions were also observed between the CMTM7 rs347134 polymorphism and AFDP in terms of LDL (*P*-interaction = 0.033), in such a way that the last quartile of the AFDP compared to the first quartile increased the terms of LDL with the GG genotype (*P* = 0.038) while having no difference for those carrying the AA/GA genotypes (*P* = 0.243) (Figure 2).

## 4. Discussion

BMI and WHtR have been widely used in obesity screening. There are international standards for determining general overweight/obesity and central obesity in adults, while for children and adolescents, different BMI and WHtR boundaries are defined by sex and age. However, due to sex, age, and regional differences, the boundaries of BMI and WHtR in different countries are not uniform. This study adopted the classification criteria for overweight and obesity screening of Chinese children and adolescents, as proposed in the guidelines for the prevention and control of overweight and obesity in Chinese children and adolescents issued by the China Working Group on Obesity of the International Life Sciences Institute (ILSI) in 2004 [24]. Current domestic studies on overweight and obesity in children are basically based on this classification criteria, so the data in this study are comparable with other relevant domestic researches.

In this study, a higher incidence of general overweight/obesity or central obesity of boys was consistent with findings among some previous works of China [4,27]. Plenty of studies have been conducted to explore and confirm the use of anthropometric indices in predicting blood pressure, glucose, and lipid profiles [28,29,30]. Results in this study also showed that general overweight/obesity or central obesity children had significantly higher SP, DP, TG, LDL, and FBG (not with central obesity), but significantly lower HDL levels compared to the normal group.

The most attractive findings of this study were that the GG genotype of CMTM7 rs347134 was related to greater odds of body weight, waist circumference, BMI, WHtR, and SP in Han Chinese male children; here shown for the first time. At present, researches on the CMTM7 gene mainly focus on cancer suppression and immune enhancement [31,32,33,34], but association or mechanistic studies between CMTM7 and obesity have not been reported anywhere in the world. There was a GWAS study that reported on a genetic locus polymorphism of the CMTM7 gene that has a significant correlation with European all-cause mortality in patients with heart failure, and can increase the risk of death from heart failure patients [35]. Adult overweight/obesity is closely related to heart failure; the increase of BMI can significantly increase the risk of heart failure [36]. So, these may provide clues for future studies on the relationship between the CMTM7 gene and obesity.

On the other hand, from an energy and nutrient perspective of traditional nutrition, it is known that some specific dietary components, including high energy intake, saturated fatty acids (SFAs), fructose, and so on, are associated with the development of obesity and other related pathologies, such as nonalcoholic fatty liver disease (NAFLD), cardiovascular disease (CVD), type 2 diabetes, etc. [37,38,39,40]. Conversely, other specific dietary components that have become more popular in recent years can prevent the development of obesity or the comorbidities associated with obesity, such as dietary fiber [41], antioxidants [42], n-3 polyunsaturated fatty acid (n-3 PUFA) [43], etc.

However, people usually have a varied diet, including all kinds of nutrients or other components among which may produce synergistic or counteracting effects. Therefore, the analysis of dietary patterns has been identified as a more realistic representation of dietary habits, since it takes into account the complex interactions between nutrients and other components of a diet, thus making interventions to change eating habits possible [44,45]. Studies on the association and interaction between SNPs and nutrient intake or dietary patterns on obesity risk factors have also made some progress [46,47], which can well find the relationship between the three, laying the foundation for further research. In our study, four dietary patterns of male children were identified, of which NSDP was related to higher FBG levels and AFDP was associated with higher levels of LDL. It was reported that the FTO gene polymorphism is significantly correlated with high protein intake and weakly correlated with low carbohydrate intake [48]. Similarly, this study revealed that male children with the GG genotype of CMTM7 rs347134 who had higher energy intake from protein but lower energy intake from carbohydrate were more inclined to AFDP. Studies showed that high protein and low carbohydrate diets have potential benefits for obesity prevention [49,50]. But in this study, the energy intake proportion between protein and carbohydrate did not meet the criteria for high-protein or low-carbohydrate diets. Furthermore, interactions were observed between the CMTM7 rs347134 polymorphism and dietary patterns in terms of LDL and FBG; that was, male children with the GG gene type displayed a significantly higher LDL in AFDP and FBG in NSDP, respectively.

The NSDP mainly consists of nuts and sweets, which need to be analyzed from two aspects. The animal experiment uncovered that a high-sucrose diet can induce glucose and lipid dysmetabolism in a Rat model [51]. A multi-ethnic sample of a children study had shown that greater fructose-rich food intake is positively associated with the TG level and negatively associated with the HDL cholesterol level [52]. Domestic research also found that fructose-rich food consumption is positively related to abdominal obesity in Chinese children [53,54]. On the contrary, nuts rich in unsaturated fatty acids (UFAs) and antioxidants, such as vitamin E, can improve the lipid profile, including the reduction of TG, TC, and LDL, as well as increase HDL [55]. Long-term nut consumption was also observed to have an inverse association with obesity but a positive association for diabetes mellitus [56]. Due to the intake of both nuts and sweets, adherence to NSDP was not found to be associated with obesity and blood lipid profiles in our study, but only had a significant correlation with higher FBG. The AFDP mainly consists of pork, fish, poultry, and vegetables, which also need to be discussed in detail. Pork is classified as red meat with high dietary SFAs; fish and poultry are classified as white meat with low SFAs. It is well known that SFAs can increase LDL. The latest study demonstrated that, compared with nonmeat as the major protein source, diets containing high amounts of either red or white meat, and without differences in other macronutrients, result in higher concentrations of LDL [57]. The effects of red and white meat are similar and observed with diets containing either low or high levels of SFAs [57]. Studies also showed that red meat intake is directly associated with risk of obesity, and higher BMI and WC [58]. On the contrary, the majority of studies are in favor of an inverse relationship between fiber-rich vegetables and weight-related outcomes [59]. It is also worth considering that some fish rich in n-3 PUFA like EPA and DHA can attenuate genetically associated long-term weight gain [60]. Owing to intake of meat, fish, and vegetables, adherence to AFDP was not found to be associated with overweight or obesity and related indexes in our study, but only had a significant correlation with higher LDL.

Additionally, although physical activity time and sleep duration were not found to be associated with obesity or its related indicators in this study, male children with the GG genotype displayed a significantly shorter sleep duration. A large sample study of 7–11-year-old Chinese children once reported that a short sleep duration is associated with obesity [61]. Another cohort study of New Zealand European children indicated that children may have a different genetic susceptibility to the effects of sleep duration on obesity [62]. We need to carry out further, large-sample studies to reveal the association between the CMTM7 rs347134 polymorphism and sleep duration on childhood obesity in the future.

It should be noted that our study has several limitations. Firstly, some scholars have proposed that BMI is a weaker predictor for relative body fat (%FAT) in children and youth [63]. Hence, further studies may take BFP (body fat percentage) and TFM (total fat mass) into account. Secondly, due to the cross-sectional design, causality cannot be inferred. More in-depth cellular and molecular biological experimental studies are needed to find out the precise mechanism. Thirdly, similar studies need to be performed on more SNPs, not only one SNP of a large genetic region as was examined in this study. The strengths of this study include: 10% of the blood samples were repeatedly experimented for confirming the results of genotyping examinations. Several confounders, such as sex, age, total energy, and physical activity, were adjusted for. Based on a database search, this is the first study to demonstrate the relationship between the CMTM7 rs347134 polymorphism and obesity, dietary patterns, sleep duration, as well as the interaction between this SNP and particular dietary patterns.

## 5. Conclusions

The present study demonstrated a relationship between a novel SNP of CMTM7 rs347134 and weight, BMI, WC, WHtR, SP, odds of general overweight/obesity, and central obesity in Han Chinese male children for the first time, and the GG genotype was a potential risk factor of obesity. Furthermore, it was revealed that male children with the GG genotype were more inclined to the AFDP and had a shorter sleep duration. Interactions were observed between this SNP and dietary patterns in terms of LDL and FBG; that was, male children with the GG gene type displayed higher LDL in AFDP and higher FBG in NSDP, respectively. The balanced dietary patterns or intake of specific foods such as nuts, vegetables, and fishes, which are rich in beneficial nutrition components, could helpfully prevent the development of obesity. All these results provide valuable clues for the prevention and intervention of childhood obesity in China.

## Figures and Tables

**Figure 1 ijerph-17-01515-f001:**
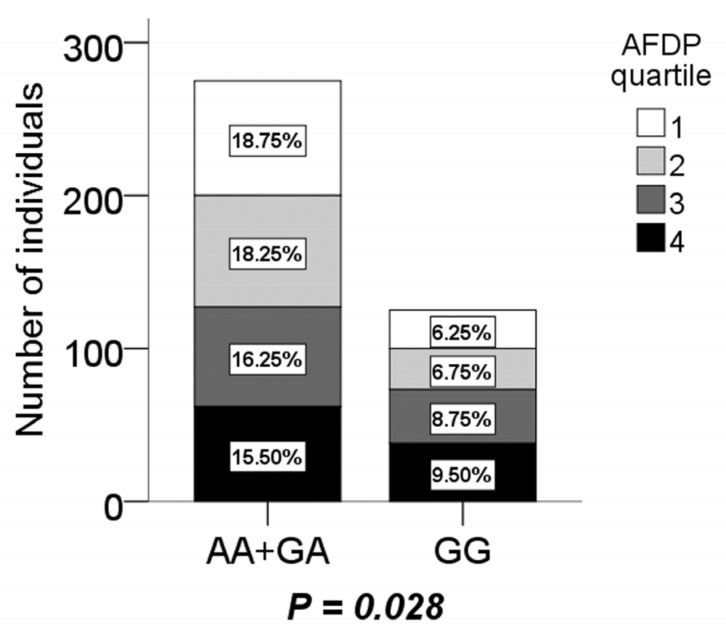
The distribution of boys with the AA + AG and GG genotype in the animal food-based dietary pattern (AFDP) quartile. The last quartile of the AFDP compared to the first quartile had a significantly higher percentage of individuals with the GG genotype, but lower with the AA + AG genotype.

**Figure 2 ijerph-17-01515-f002:**
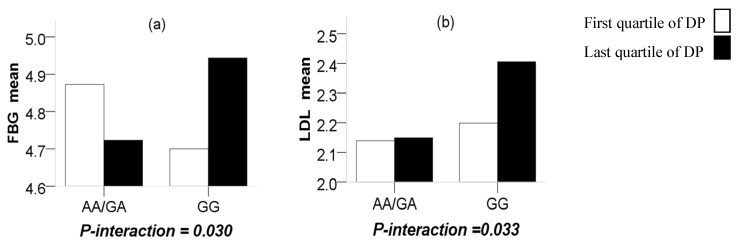
(**a**) The interaction (*P* = 0.030) between the CMTM7 rs347134 polymorphism and the nuts and sweets-based dietary pattern (NSDP) on FBG (*P* = 0.040 and 0.031 between the first and last quartile in AA/GA and GG, respectively); (**b**) the interaction (*P* = 0.033) between the CMTM7 rs347134 polymorphism and the AFDP on TG (*P* = 0.243 and 0.038 between the first and last quartile in AA/GA and GG, respectively).

**Table 1 ijerph-17-01515-t001:** Obesity-related variables compared by general overweight/obesity, central obesity, and normal.

Variables	Normal (*n* = 1041)	General Overweight/Obesity (*n* = 251)	*p*-Value ^1^	Normal (*n* = 1104)	Central Obesity (*n* = 188)	*p*-Value ^1^
SP (mmHg)	97.87 ± 9.13	104.14 ± 9.95	<0.001	98.16 ± 9.20	104.52 ± 10.21	<0.001
DP (mmHg)	60.83 ± 6.55	63.78 ± 6.86	<0.001	60.98 ± 6.58	63.89 ± 6.97	<0.001
TG (mmol/L)	0.81 ± 0.48	0.92 ± 0.58	0.002	0.81 ± 0.49	0.97 ± 0.51	<0.001
TCH (mmol/L)	3.85 ± 0.82	3.96 ± 0.68	0.063	3.86 ± 0.81	3.97 ± 0.73	0.077
HDL (mmol/L)	1.43 ± 0.27	1.34 ± 0.24	<0.001	1.43 ± 0.27	1.31 ± 0.24	<0.001
LDL (mmol/L)	2.27 ± 0.64	2.41 ± 0.60	0.001	2.26 ± 0.64	2.49 ± 0.58	<0.001
FBG (mmol/L)	4.63 ± 0.45	4.72 ± 0.42	0.005	4.64 ± 0.45	4.69 ± 0.42	0.138

Note: Variables are mean value ± SD (standard deviation); (n) population size. ^1^ Independent *t*-test.

**Table 2 ijerph-17-01515-t002:** Obesity-related variables compared by sex.

Variables	Male (*n* = 658)	Female (*n* = 634)	*p*-Value
General overweight/obesity rate	24.17%	14.51%	<0.001 ^a^
Central obesity rate	16.72%	12.30%	0.024 ^a^
BMI	17.08 ± 2.85	16.19 ± 2.41	<0.001 ^b^
Waist(cm)	57.24 ± 8.03	54.05 ± 6.60	<0.001 ^b^
WHtR	0.43 ± 0.05	0.41 ± 0.04	<0.001 ^b^
SP (mmHg)	100.05 ±9.72	98.09 ± 9.41	<0.001 ^b^
DP (mmHg)	61.70 ± 6.76	61.09 ± 6.65	0.098 ^b^
TCH (mmol/L)	3.85 ± 0.77	3.89 ± 0.82	0.351 ^b^
TG (mmol/L)	0.82 ± 0.53	0.85 ± 0.47	0.215 ^b^
LDL (mmol/L)	2.26 ± 0.61	2.33 ± 0.66	0.039 ^b^
HDL (mmol/L)	1.44 ± 0.28	1.39 ± 0.26	0.002 ^b^
FBG (mmol/L)	4.72 ± 0.46	4.58 ± 0.42	<0.001 ^b^

Note: continuous variables are mean value ± SD; (n) population size. ^a^ Chi-squared test. ^b^ Independent *t*-test.

**Table 3 ijerph-17-01515-t003:** Obesity-related variables according to the CMTM7 rs347134 genotype for male children.

Variables	AA + GA (*n* = 465)	GG (*n* = 193)	*P* ^1^	*P*-Ancova ^2^
Weight (kg)	32.04 ± 7.59	34.09 ± 8.45	0.016	0.074
BMI	17.20 ± 2.77	17.98 ± 3.30	0.015	0.039
Waist (cm)	58.04 ± 7.77	60.12 ± 8.96	0.019	0.046
WHtR	0.43 ± 0.05	0.44 ± 0.06	0.039	0.047
SP (mmHg)	99.45 ± 9.61	101.81 ± 9.77	0.024	0.049
DP (mmHg)	61.72 ± 6.39	62.49 ± 6.20	0.259	0.347
TG (mmol/L)	0.81 ± 0.52	0.87 ± 0.68	0.362	0.448
TCH (mmol/L)	3.87 ± 0.71	3.75 ± 0.90	0.179	0.162
HDL (mmol/L)	1.45 ± 0.27	1.42 ± 0.29	0.331	0.343
LDL (mmol/L)	2.26 ± 0.59	2.24 ± 0.64	0.792	0.734
FBG (mmol/L)	4.75 ± 0.43	4.71 ± 0.46	0.466	0.286

Note: variables are mean value ± SD; (n) population size. ^1^ For the crude ANOVA model. ^2^ For the adjusted ANCOVA model by age, total energy intake, and physical activity.

**Table 4 ijerph-17-01515-t004:** Factor loadings for the four identified dietary patterns of the male children.

Food Groups	Dietary Patterns			
	HBDP	NSDP	AFDP	WDDP
Vegetables	0.632		0.412	
Eggs	0.609			
Tubers	0.593			
Dairy	0.334			0.388
Nuts		0.713		
Pastries		0.704		
Sweets		0.457		
Pork			0.703	
Fish			0.696	
Poultry			0.376	
Wheaten				0.765
Rice				−0.618
Eigenvalues	1.673	1.513	1.462	1.429
Variance (%)	8.0	7.2	7.0	6.8
Total variance (%) = 32.4				

Note: values are factor loadings of dietary patterns measured by factor analysis. Factor loadings below ± 0.3 are not shown.

**Table 5 ijerph-17-01515-t005:** Obesity-related indices of male children by quartile of dietary patterns.

	Quartile 1	Quartile 2	Quartile 3	Quartile 4	*P* ^3^	*P*-Ancova ^4^
Overweight/obesity incidence rate (%) ^1^						
HBDP	29.1	27.2	21.4	22.3	0.51	
NSDP	27.2	23.3	20.4	29.1	0.47	
AFDP	22.3	27.2	26.2	24.3	0.86	
WDDP	26.2	25.2	23.3	25.2	0.97	
Central obesity incidence rate (%) ^1^						
HBDP	25.3	33.3	21.3	20.0	0.26	
NSDP	28.0	29.3	20.0	22.7	0.54	
AFDP	21.3	28.0	29.3	21.3	0.57	
WDDP	24.0	29.3	20.0	26.7	0.63	
Weight (kg) ^2^						
HBDP	32.83 ± 7.57	32.85 ± 8.08	32.37 ± 8.24	32.68 ± 7.87	0.97	0.97
NSDP	34.31 ± 8.50	31.92 ± 7.64	32.09 ± 7.15	32.40 ± 8.19	0.12	0.68
AFDP	31.27 ± 7.01	32.80 ± 8.09	33.84 ± 8.76	32.82 ±7.61	0.15	0.26
WDDP	32.61 ± 7.92	32.87 ± 8.00	32.13 ± 8.03	33.12 ± 7.80	0.84	0.80
BMI ^2^						
HBDP	17.52 ± 2.78	17.52 ± 3.22	17.29 ± 3.03	17.45 ± 2.85	0.94	0.94
NSDP	17.77 ± 3.15	17.34 ± 3.01	17.16 ± 2.66	17.51 ± 3.03	0.52	0.69
AFDP	16.87 ± 2.48	17.64 ± 3.33	17.79 ± 3.22	17.48 ±2.72	0.14	0.19
WDDP	17.50 ± 3.05	17.52 ± 3.09	17.22 ± 2.81	17.55 ±2.93	0.85	0.86
Waist (cm) ^2^						
HBDP	59.12 ± 7.66	59.02 ± 8.83	58.22 ± 8.73	58.42 ± 7.62	0.83	0.85
NSDP	59.88 ± 8.66	58.13 ± 7.98	58.15 ± 7.78	58.61 ± 8.38	0.39	0.77
AFDP	57.22 ± 7.03	58.92 ± 8.72	60.03 ± 9.20	58.61 ± 7.57	0.11	0.19
WDDP	58.76 ± 8.24	59.36 ± 8.66	57.87 ± 7.87	58.79 ± 8.09	0.64	0.55
WHtR ^2^						
HBDP	0.43 ± 0.05	0.43 ± 0.06	0.43 ± 0.05	0.43 ± 0.05	0.81	0.83
NSDP	0.43 ± 0.05	0.43 ± 0.05	0.43 ± 0.05	0.43 ± 0.05	0.81	0.79
AFDP	0.42 ± 0.04	0.43 ± 0.06	0.44 ± 0.06	0.43 ± 0.05	0.20	0.22
WDDP	0.43 ± 0.05	0.44 ± 0.05	0.43 ± 0.05	0.43 ± 0.05	0.58	0.59
SBP (mmHg) ^2^						
HBDP	100.07 ± 8.99	100.88 ± 10.96	98.03 ± 8.58	101.76 ± 9.88	0.14	0.14
NSDP	100.49 ± 9.55	99.34 ± 9.70	99.96 ± 9.41	100.95 ± 10.22	0.68	0.72
AFDP	99.61 ± 9.12	98.45 ± 9.34	101.39 ± 10.16	101.29 ± 10.01	0.19	0.17
WDDP	100.87 ± 10.10	101.02 ± 10.26	98.49 ± 9.36	100.36 ± 8.99	0.23	0.37
DBP (mmHg) ^2^						
HBDP	61.64 ± 6.18	62.76 ± 6.82	61.00 ± 6.23)	62.43 ± 6.01	0.20	0.21
NSDP	62.16 ± 5.79	61.65 ± 6.65	61.18 ± 6.48)	62.84 ± 6.35	0.29	0.32
AFDP	62.19 ± 6.19	61.17 ± 6.74	62.23 ± 6.24)	62.24 ± 6.18	0.56	0.67
WDDP	62.26 ± 6.30	62.38 ± 5.76	60.88 ± 6.55)	62.31 ± 6.66	0.28	0.29
TG (mmol/L) ^2^						
HBDP	0.77 ± 0.44	0.80 ± 0.52	0.80 ± 0.51	0.95 ± 0.76	0.12	0.68
NSDP	0.78 ± 0.45	0.78 ± 0.54	0.80 ± 0.52	0.95 ± 0.73	0.11	0.18
AFDP	0.86 ± 0.51	0.71 ± 0.45	0.86 ± 0.66	0.88 ± 0.64	0.11	0.37
WDDP	0.83 ± 0.68	0.90 ± 0.59	0.72 ± 0.45	0.86 ± 0.54	0.12	0.13
TCH (mmol/L) ^2^						
HBDP	3.87 ± 0.70	3.76 ± 0.71	3.91 ± 0.81	3.78 ± 0.87	0.48	0.44
NSDP	3.86 ± 0.78	3.89 ± 0.61	3.85 ± 0.81	3.72 ± 0.88	0.43	0.91
AFDP	3.76 ± 0.76	3.82 ± 0.71	3.92 ± 0.84	3.82 ± 0.79	0.57	0.37
WDDP	3.89 ± 0.69	3.81 ± 0.82	3.91 ± 0.78	3.71 ± 0.81	0.25	0.58
HDL (mmol/L) ^2^						
HBDP	1.40 ± 0.25	1.44 ± 0.27	1.45 ± 0.28	1.46 ± 0.30	0.47	0.29
NSDP	1.41 ± 0.28	1.44 ± 0.29	1.45 ± 0.26	1.46 ± 0.27	0.69	0.55
AFDP	1.45 ± 0.26	1.42 ± 0.27	1.45 ± 0.29	1.44 ± 0.29	0.79	0.63
WDDP	1.44 ± 0.27	1.41 ± 0.25	1.47 ± 0.30	1.43 ± 0.27	0.50	0.47
LDL (mmol/L) ^2^						
HBDP	2.26 ± 0.57	2.20 ± 0.59	2.31 ± 0.66	2.23 ± 0.60	0.62	0.61
NSDP	2.28 ± 0.67	2.23 ± 0.53	2.28 ± 0.63	2.22 ± 0.59	0.85	0.92
AFDP	2.18 ± 0.51	2.16 ± 0.61	2.36 ± 0.65	2.37 ± 0.63	0.05	0.04
WDDP	2.26 ± 0.66	2.34 ± 0.57	2.22 ± 0.64	2.19 ± 0.55	0.32	0.34
FBG (mmol/L) ^2^						
HBDP	4.80 ± 0.41	4.71 ± 0.44	4.73 ± 0.42	4.71 ± 0.48	0.47	0.48
NSDP	4.61 ± 0.43	4.68 ± 0.46	4.69 ± 0.43	4.77 ± 0.43	0.03	0.04
AFDP	4.73 ± 0.42	4.73 ± 0.41	4.72 ± 0.47	4.78 ± 0.45	0.80	0.70
WDDP	4.73 ± 0.41	4.70 ± 0.47	4.76 ± 0.41	4.66 ± 0.44	0.13	0.11

^1^ Categorical variables were compared using a Chi-squared test. ^2^ Continuous variables are mean value ± SD. ^3^ For the crude ANOVA model. ^4^ For the adjusted ANCOVA model by age and physical activity.

**Table 6 ijerph-17-01515-t006:** Energy intake, physical activity, and sleep duration compared by genotypes in boys.

Variable	AA + GA (*n* = 465)	GG (*n* = 193)	*p*-Value ^1^
Total energy intake (kcal/d)	2074.47 ± 474.65	2091.57 ± 559.29	0.856
Energy from protein (%)	22.03 ± 5.40	24.05 ± 5.45	0.043
Energy from fat (%)	29.09 ± 9.43	29.95 ± 9.37	0.428
Energy from carbohydrate (%)	54.08 ± 10.64	51.16 ± 10.91	0.046
Physical activity time (h/d)	2.52 ± 1.39	2.57 ± 1.42	0.766
Sedentary time (h/d)	4.25 ± 1.28	4.21 ± 1.37	0.843
Sleep duration (h/d)	8.85 ± 1.08	8.53 ± 1.09	0.031

^1^ Independent *t*-test.

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
