# Peer review of "Interaction of the CMTM7 rs347134 Polymorphism with Dietary Patterns and the Risk of Obesity in Han Chinese Male Children"

_ijerph, 2020, doi:10.3390/ijerph17051515_

Round 1

Reviewer 1 Report

Comments to Author

I enjoyed reading this paper and found the work to be of important value. The complexity of obesity and its connection to childhood factors is an important area of investigation. The interplay between genetic and environmental factors in the etiology of obesity development will provide an opportunity to clarify the differences in incidence.

The paper describes a strong methodological assessment of genetic markers and their connection to multiple environmental antecedents of obesity (dietary patterns, physical activity, sleep).

The approach provides a logical process for the reader to follow.  Errors in language and grammar interfere with the communication of the message and would benefit from editing.

Major comments:

No major concerns

Minor comments:

Line 33: should ‘obesity’ be ‘obese’

Line 40: phrase ‘not a single role’ is not clear. Should it be ‘no single path’?

Line 40: the term genders used incorrectly – consider the term ‘sex’ (sex – biological attributes; gender – social constructed concepts)

Line 44: meaning of phrase ‘well known mainly include’ is unclear in this sentence

Line 45: remove ‘and so on’

Line 47: should a citation accompany the list of genes? The phrase ‘et al.’ indicates a list of authors but no author names have been included

Line 59: in the previous paragraph obesity in ‘Chinese children’ was discussed; now in line 59 the population is Han Chinese children. Consider providing a small description to contextualize this population

Line 60: should ‘were’ be ‘is’ considering the ‘is’ in Line 64? Verb tense makes the meaning slightly difficult to understand

Line 67: should author’s (meaning one author’s knowledge) be authors’ knowledge (meaning multiple authors’ knowledge) – consider rephrasing first part of sentence – To our knowledge, …..

Line 72: not a full sentence

Line 81: “Lastly, 30 ~ 40 pupils were drew from the classes as the research objects.” Grammar ; should ‘objects’ be participants?

Line 82: what year in grades or age of the children were sampled? General information, as exact descriptive statistics around age are provided in results. Basic group information would be useful.

Line 83 – 334: Continue with small grammatical, sentence, and word editing throughout the submission

Reviewer 2 Report

The proposed manuscript by Zhu and co-authors presents the results from a cross-sectional study aiming to investigate the association of CMTM7 gene rs347134 single nucleotide polymorphism (SNP) with obesity, and its interaction with dietary patterns in Han Chinese children. This study was a multi-centered randomized cluster controlled trial and part of a subordinate of a nutrition-based comprehensive intervention study on childhood obesity in China (NISCOC) and involved a total of 1,292 boys. The results from this study indicate that GG genotype of CMTM7 rs347134 is potentially a novel obesity risk factor for Han Chinese male children associated with dietary patterns.

I carefully reviewed the manuscript and did not find any issues with the study design, data analysis or data representation. Manuscript is written well and no English editing is needed either.

Author Response

Thank you very much for your appreciation and recognition of this study. 

Best Regards

Reviewer 3 Report

The manuscript is interesting and shows important information (public and clinical health). The manuscript has an adequate structure. The wording is good. The methodology used is sufficient and consistent with the objective of the study. The information presented in the manuscript will be of interest to researchers, health professionals and health authorities. However I have the following comments.

Major Comments:
1. I suggest including a brief paragraph regarding which specific dietary components would be associated with the development of obesity and other pathologies such as NAFLD in children. Such as, high energy intake, saturated fatty acids and fructose (corn syrup).

Suggested reference:

Hernandez-Rodas et al., Relevant Aspects of Nutritional and Dietary Interventions in Non-Alcoholic Fatty Liver Disease. Int J Mol Sci. 2015; 16: 25168-98.

2. The discussion is good and is based on the results. In this regard, considering the results "food groups". Again I suggest deepening the discussion. Writing a brief paragraph, regarding components that could prevent the development of obesity or the comorbidities associated with obesity. Such as: dietary fiber, antioxidants present in food, n-3 PUFA, etc.

Suggested reference:

Hernandez-Rodas et al., Relevant Aspects of Nutritional and Dietary Interventions in Non-Alcoholic Fatty Liver Disease. Int J Mol Sci. 2015; 16: 25168-98.

3. Only one question (it is not necessary to include the answer in the manuscript): Considering the results, is it possible to project the observed results to an increase in the development of chronic pathologies in adult life?

Minor comments:
1. Is reference 19 necessary?
2. Improve the wording of the study objective
3. Improve resolution (graphic quality) of figures 1 and 2.

Author Response

Point 1: I suggest including a brief paragraph regarding which specific dietary components would be associated with the development of obesity and other pathologies such as NAFLD in children. Such as, high energy intake, saturated fatty acids and fructose (corn syrup).

Suggested reference:

Hernandez-Rodas et al., Relevant Aspects of Nutritional and Dietary Interventions in Non-Alcoholic Fatty Liver Disease. Int J Mol Sci. 2015; 16: 25168-98.

Response 1: Thanks a lot for your suggestion. We really ignored the discussion about specific nutrients or other dietary components associated with obesity and other related pathologies. We have read the reference you suggested carefully and searched the other related literatures, subsequently supplemented the relevant content in the manuscript in red.

Point 2: The discussion is good and is based on the results. In this regard, considering the results "food groups". Again I suggest deepening the discussion. Writing a brief paragraph, regarding components that could prevent the development of obesity or the comorbidities associated with obesity. Such as: dietary fiber, antioxidants present in food, n-3 PUFA, etc.

Suggested reference:

Hernandez-Rodas et al., Relevant Aspects of Nutritional and Dietary Interventions in Non-Alcoholic Fatty Liver Disease. Int J Mol Sci. 2015; 16: 25168-98.

Response 2: These nutrition components are indeed the focus of research in recent years. We have supplemented and deepened the discussion about relevant content in the manuscript in red. Thank you again for suggestion.

Point 3: Only one question (it is not necessary to include the answer in the manuscript): Considering the results, is it possible to project the observed results to an increase in the development of chronic pathologies in adult life?

Response 3: It’s really a very good question. Since the objects of this study were children, such a projection was not made in the manuscript for scientific rigor. In fact, the dietary patterns of children in the same region are similar to those of adults, and also influenced by adult eating habits. According to the results of this study, the effects of children's food composition on obesity related indicators are consistent with many mainstream views in the field of nutrition about adult chronic diseases including NAFLD, cardiovascular disease, diabetes, etc. Therefore, we believe that the results of this study may provide certain reference significance for the dietary guidance of prevention and intervention of obesity related chronic pathologies in adult life.

Point 4: Is reference 19 necessary?

Response 4: Actually, it is not necessary and has been removed.

Point 5: Improve the wording of the study objective

Response 5: The wording of the study objective has been improved at the end of introduction.

Point 6: Improve resolution (graphic quality) of figures 1 and 2.

Response 6: Resolution of figures 1 and 2 has been improved to 600 dpi and saved as TIFF format.

Round 2

Reviewer 3 Report

The authors made all changes suggested. The manuscript can be accepted for publication.